# Evaluating Johnson and Johnson COVID-19 Vaccination Outcomes in a Low-Income Hispanic Population

**DOI:** 10.3390/vaccines11010148

**Published:** 2023-01-10

**Authors:** Christopher Lamb, Christopher Owens, Wendy Gamboa, Alfredo Lopez-Yunez

**Affiliations:** 1Weatherhead School of Management, Case Western Reserve University, Cleveland, OH 44106, USA; 2Alivio Medical Center, Indianapolis, IN 46219, USA

**Keywords:** Johnson & Johnson COVID-19 vaccine, COVID-19 antibodies, COVID-19 humoral response, Clungene^®^, COVID-19 Hispanic impact, COVID-19 low-income population

## Abstract

Background: A pilot study was performed at a low-income emergency care clinic to assess the humoral immune response to the Johnson & Johnson (J&J) COVID-19 vaccine (Ad26.COV2.S) to better understand how to evaluate the COVID-19 health status of its Hispanic patient population following vaccination. Methods: This study used the Clungene^®^ SARS-CoV-2 IgG/IgM Rapid Test Cassette to determine the presence of binding antibodies resulting from the J&J COVID-19 vaccine. The Clungene test principle is based on the receptor-binding domain (RBD) of the spike protein. Antibodies targeting the spike protein are considered an appropriate measure of humoral response from spike-based vaccines. Results: The study confirmed previous research that antibodies wane over time, and results are consistent with reported vaccine efficacy. There was a statistically significant relationship between the humoral immune response and demographic and health status variables. Conclusions: COVID-19 negative patients can be easily and efficiently monitored to determine the success and durability of COVID-19 vaccines in low-income minority populations. The use of simple low-cost spike targeted COVID-19 antibody lateral flow devices may serve as a useful adjunct to assist community-based physicians on the COVID-19 health status of its patients. Further research is needed to confirm the utility of this approach.

## 1. Introduction

SARS-CoV-2, designated COVID-19 by the World Health Organization, caused a global pandemic that severely disrupted all spheres of health care delivery [1,2,3,4,5,6]. The death toll of COVID-19 is 6.62 million people as of 21 November 2022 [7]. However, new studies have found that the death toll is 18.2 million people or three times previous reports confirming its status as the greatest public health catastrophe since the 1918 influenza pandemic which killed 50 million people [8,9,10].

As multiple vaccines have been developed with more than 12.97 billion doses administered globally to over 5.46 billion people or 68.4% of the world’s population [11,12,13], the pandemic has receded and has been downgraded to endemic status in countries with high vaccination rates [14,15,16,17,18]. Whether or not this change is warranted, policy makers are lifting pandemic restrictions related to masks and social distancing [19]. The change is predicated on two assumptions: that the percent of the population has sufficient immunity either from vaccination or prior infection and the severity of recent variants is significantly lower in terms of hospitalizations and a non-zero “normal” mortality [20,21]. However, as the COVID-19 virus continues to mutate and variants cause new surges in the United States and Europe, vaccine effectiveness wanes due to reduced antibody neutralization, necessitating the need for better public health tools to check the contagion [22,23].

Globally there are more than 34 COVID-19 vaccines that have been granted regulatory approval, of which 10 have been recognized by the World Health Organization [24,25]. COVID-19 vaccine efficacy from large-scale observational studies generally shows how well vaccination protects people against outcomes such as infection, symptomatic illness, hospitalization, and death. Vaccine studies attempt to estimate individual protection under “real-world” conditions [26] at a given point in time. While COVID-19 vaccines have demonstrated a significant reduction in hospitalizations and infection [27,28,29,30], effectiveness declines within 4–6 months, thereby supporting the public health recommendation for booster shots [31]. However, surges in COVID-19 infections are increasingly caused by subvariants in populations with high rates of vaccination [32].

While immunity acquired through vaccination has been extensively studied and demonstrated. the underlying mechanism of action is not completely understood. It is assumed that the response to the COVID-19 vaccine elicits an adaptive immune response: both an initial neutralizing antibody response as well as providing a memory that can generate virus-specific T-cell or B-lymphocytes if there is subsequent antigen exposure [33]. This response initially appeared to be robust against emerging COVID-19 variants as well but recent reports suggest otherwise [34,35].

Neutralizing antibodies wane within 6 months, with questionable efficacy with new variants, and afterwards, reliance on memory cells is required, likely maintained but diminishing for 12 months following infection [36,37]. An antibody response can be measured, memory cannot. The strength or efficacy of the memory has yet to been determined as it likely varies by individual circumstances, including degree of a compromised immune system or impairment by disease. Ways in which to apply this knowledge during uncertainty in everyday office-based clinical practice are needed. Given new estimates from the US CDC showing the rapidly spreading Omicron subvariant known as BA.5 becoming dominant among new coronavirus cases, the need for new cost-effective tools, particularly in high-risk low-income minority communities, is urgent [38].

A pilot study was performed at an emergency care clinic to assess the humoral immune response to the Johnson & Johnson (J&J) COVID-19 vaccine (Ad26.COV2.S) to better understand how to best serve its patient population [29]. Alivio Medical Center is a non-affiliated, free-standing, point-of-care medical center in Indianapolis Indiana serving a low-income Hispanic community [39]. There are limited studies which attempt to measure the challenges to implement vaccination and track the progress in low-income minority communities. Further, Hispanic and other minority populations are underrepresented in the published literature despite data that show a disparate COVID-19 impact [40,41,42]. For reasons not fully understood, the Hispanic population has had worse outcomes than any other racial group, including higher infection rates and outsized death rates based on weighted population distribution [43,44,45,46]. Although the extant literature is primarily descriptive, our prior work and other sources suggest that the excess risk may be related to work exposure and disproportionately affecting Hispanics who are more likely unable to work from home [44,47,48]. Therefore, studies involving minority persons are crucial for hypothesis generation and testing to develop health policy and treatment strategies which make outcomes more equitable. The present study provides further insights into the specific characteristics of the application and study of COVID-19 vaccination to a low-income Hispanic community.

### Background

On 29 March 2021, the J&J COVID-19 vaccine was authorized for use under an Emergency Use Authorization (EUA 27205/56) for active immunization to prevent Coronavirus Disease 2019 (COVID-19) caused by severe acute respiratory syndrome coronavirus 2 (SARS-CoV-2). The Alivio clinical team previously studied the impact of COVID-19 on its low-income Hispanic population prior to the availability of vaccines [48]. Following approval of the J&J COVID-19 vaccine, Alivio initiated a follow-up study to assess the impact of vaccination on its patient population.

COVID-19 lateral flow assays can be used to distinguish antibodies from infection or vaccination; however, the identity of target SARS-CoV-2 antigens in the LFAs is critical. J&J uses the S antigen. In populations that receive spike-based vaccines, LFAs targeting the N and S antigens can be used to identify natural or vaccine-induced SARS-CoV-2 antibodies [49].

The Alivio study team used the Clungene^®^ SARS-CoV-2 IgG/IgM Rapid Test Cassettes to determine the presence of binding antibodies resulting from the J&J vaccine. The Clungene^®^ test principle is based on the receptor-binding domain (RBD) of both the spike and nucleocapsid proteins [50]. Antibodies targeting the spike protein are considered a better measure of humoral response [30]. Therefore COVID-19 negative patients at the start of the study could be easily monitored to determine the success and durability of the vaccine in this study population following vaccination. 

The J&J is single-shot vaccine. In the United States, stable vaccine effectiveness of 76 percent (95% CI, 75–77%) for COVID-19-related infections and 81 percent (95% CI, 78–82%) for COVID-19-related hospitalizations [51]. The J&J/Janssen COVID-19 vaccine contains a vector virus that instructs cells in the body to create an immune response intended to protect recipients from getting sick with COVID-19 in the future [52]. Recent data show the J&J vaccines are as effective as other FDA authorized vaccines [53].

Because of initially reported negative data regarding both safety and efficacy, only a small percentage of total vaccines administered have been J&J—approximately 37 million doses in the United States and Europe out of a total of 1.4 billion [54]. Recent data show an overall decrease in vaccine effectiveness between months 1–6 [28,55]

A booster at two months showed that a second shot of the J&J COVID-19 vaccine given 56 days after the first provided 100% protection (CI, 33–100%) against severe/critical COVID-19–at least 14 days post-final vaccination; 75% protection against symptomatic (moderate to severe/critical) COVID-19 globally (CI, 55–87%); and 94% protection against symptomatic (moderate to severe/critical) COVID-19 in the United States (CI, 58–100%). With this booster given two months after the first shot, antibody levels rose to four to six times higher than those observed after the single shot. A booster shot at six months provided a 12-fold increase in antibodies [51,56].

## 2. Materials and Methods

### 2.1. Study Design

This was a prospective, longitudinal observational study conducted using a pre-defined protocol and cohort. The study was conducted under a Western IRB approval (sponsor: Alivio Medical Center, Indianapolis, IN, USA; IRB Pr #: 20210840).

### 2.2. Protocol and Study Population

The kinetics of SARS-CoV-2 anti-spike (S) IgG antibodies were analyzed up to six months after a single-dose vaccination against COVID-19 and the durability of the humoral response for the infection-naïve patients was compared. Serum samples for anti-S antibodies were obtained 7 to 64 days after a dose of the J&J vaccine, and again after 90 days. Patients consented to participate in a study that would track their antibodies following vaccination. Eligible subjects were COVID-19 antibody negative using a commercially available assay. Patients were asked to return twice over a period of 120 days following vaccination and tested for COVID-19 antibodies. Subjects were low-income Hispanic patients of the Alivio Medical Center who voluntarily presented for a routine vaccination. The study population included a cohort of Hispanic patients presenting to the Alivio center during the COVID-19 pandemic.

### 2.3. Methods

Patients entering Alivio Medical Center between 8 June 2021 and 25 October 2021, who were seeking a COVID-19 vaccination were evaluated for SARS-CoV-2 infection using a medical questionnaire for symptoms suspicious for COVID-19 administered by a trained health care professional. Patients with no history of infection were tested for antibodies. Those with both a negative history and negative antibodies were invited to enroll in the study. Qualified consenting subjects had their serum tested after the dose of the J&J COVID-19 vaccine and after 3 months and 6 months. Patients with a positive antibody test prior to vaccination were excluded. All study staff in this investigation spoke Spanish fluently and conducted interviews, collected demographic and socioeconomic information, and provided treatment in the Spanish language. 

Electronic medical records in iSalus were used to compile patients’ baseline demographics, comorbidities, medications, laboratory, and socioeconomic data. Chest radiographs were obtained and the clinical course was documented. Socioeconomic data such as employment status, time to recovery, and missed work were collected by self-report in response to an interview by our study staff.

Subjects selected for the detection of SARS-CoV-2 antibodies (IgM and IgG) were tested using the point-of-care CLUNGENE^®^ SARS-CoV-2 VIRUS IgM/IgG Rapid Test Cassette lateral flow immunoassay (LFA). The CLUNGENE^®^ Test was commercially available in the United States under an FDA-approved Emergency Use Authorization (EUA201121) and Europe (CE Mark reference 02PBJ267 dated 9 March 2020) [57]. The CLUNGENE^®^ Test has been previously studied, including the use of the test in the offices of general practitioners, evaluating the presence of antibodies in convalescent plasma donors, and how the test performs at a point-of-care facility [58,59,60,61]. This LFA can detect specific SARS-CoV-2 antibodies and differentiate between IgG and IgM antibody classes [50,62]. Previously, the manufacturer of the LFA (Hangzhou Clongene Biotech Co., Ltd., Hangzhou, China) validated this immunoassay for the qualitative detection of antibodies to SARS-CoV-2 and the data were submitted to the US FDA consistent with requirements of the Emergency Use Authorization [57,63,64]. The test employs one drop (~10 µL) of whole blood added to a well followed by two drops of buffer. Results are available in 15 min. According to the instructions for use (IFU), the sensitivity was 87%, the specificity was 98% yielding a test accuracy of 93%. For the present analysis, a test was considered positive if IgM, IgG, or both antibodies were present. 

A correlation analysis was performed on the patient data. Moderate to strong Pearson R’s (<|0.3|) were reported, particularly those in correlation to the antibody results. Any patterns and observations are provided in the results.

## 3. Results

Our results are derived from the 39 participants returned for at least one (1) follow-up appointment to test for an antibody response; 74 participants were requested to participate, 53 qualified to participate, 39 returned for at least one (1) follow-up appointment to test for an antibody response (See Table 1 and Table 2). Of the 39 participants, 16 (41%) tested negative for antibodies in one or more follow-up appointments; 5 (12%) reported negative in both appointments, 10 (26%) reported for one appointment, and 1 (3%) was negative then had become positive. The remaining 23 (59%) participants tested positive during one or both appointments; 15 (38%) reported positive in both appointments and 8 (21%) reported for one appointment.

There is a trending correlation between antibody response and the number of days after vaccination. Although the correlation is weak when observing the total dataset and the two follow-up appointments, there is a clear shift towards a negative correlation (<−0.3) with each passing month. The correlation is positive, yet weak in the first month (0.21), but trends downwards starting in the second month; correlations are −0.31, −0.23, −0.44, and −0.37, respectively (See Table 3).

### 3.1. All Visits (60 Visits Total; 39 Individual Patients)

Patients with positive antibody results are more likely to be smaller, younger women with no fatigue or previous COVID infection, and were likely to attend a follow-up visit 2–3 months after vaccination. The correlations for all visits (60 visits total; 39 individual patients) summarize consistent patterns in the data as we drill down. There is a positive correlation between a positive antibody result and women (0.15). There is a negative correlation between a positive antibody result and men (−0.15), age (−0.12), previous COVID infection (−0.18), height (−0.14), weight (−0.13), fatigue (−0.23), and days after vaccination (−0.22).

### 3.2. Visits Less Than 30 Days after Vaccination (N = 15 Visits)

For the subgroup of patients who visited less than 30 days after vaccination (N = 15 visits; 15 patients), there is a positive correlation between a positive antibody result and women (0.34), pain at injection site (0.12), and days after vaccination (0.21). There is a negative correlation between a positive antibody result and age (−0.4), diabetes (−0.21), hypertension (−0.45), hyperlipidemia (−0.21), height (−0.35), weight (−0.55), headache (−0.12), fatigue (−0.85), muscle aches (−0.34), nausea (−0.44), dizziness (−0.21), fever (−0.21), and duration of symptoms (−0.26). 

### 3.3. Visits 30–60 Days after Vaccination (N = 6)

There is a positive correlation between a positive antibody result and men (0.45), diabetes (0.2), hypertension (0.2), rheumatological diseases (0.2), weight (0.34), pain at injection site (0.32), nausea (0.2), and dizziness (0.2). There is a negative correlation between a positive antibody result and height (−0.15), fatigue (−1), and days after vaccination (−0.31).

### 3.4. Visits 60–90 Days after Vaccination (N = 9)

There is a positive correlation between a positive antibody result and men (0.19), age (0.15), headache (0.25), fatigue (0.38), muscle aches (0.25), nausea (0.25), and duration of symptoms (0.32). There is a negative correlation between a positive antibody result and and rheumatological diseases (−0.5), pain at injection site (−0.5), and days after vaccination (−0.23).

### 3.5. Visits 90–120 Days after Vaccination (N = 14)

There is a positive correlation between a positive antibody result and diabetes (0.24) and hyperlipidemia (0.24). There is a negative correlation between a positive antibody result and men (−0.29), age (−0.24), height (−0.32), weight (−0.27), and days after vaccination (−0.44).

### 3.6. Visits More Than 120 Days after Vaccination (N = 16)

There is a positive correlation between a positive antibody result and women (−0.22) and hypertension (0.10). There is a negative correlation between a positive antibody result and previous COVID infection (−0.29), diabetes (−0.19), hyperlipidemia (−0.20), height (−0.21), weight (−0.15), pain at injection site (−0.29), duration (−0.29), and days after vaccination (−0.33).

## 4. Discussion

Data generated in this study suggest that the administration of the J&J vaccine produces a detectable antibody response in just 14% of Hispanic subjects by a single injection. Over the course of the study, the majority of these 10 patients had a detectable antibody response through to 120 days, while there are several significantly positive correlations between patient factors and positive antibody response at various time points. This and other socioeconomic factors suggest Hispanic patients are particularly vulnerable to COVID-19. However, data from larger studies have inconsistent results that both support and contradict our observation. In Shay et al., Hispanic patients taking the Janssen vaccine are the least likely to have adverse events (2.2%) when compared to Asian (2.6%), White (2.7%), and Black (5.6%) patient samples; this contradicts our claim [65]. In Xu et al., their sample of Hispanic patients taking the J&J vaccine had a worse standardized mortality rate than the other major populations, except the Black patients [66]. Further research is required to fill the knowledge gap of COVID-19 vaccine effectiveness and adverse reactions between ethnicity groups; filling the gap will help HCPs navigate the brand nuances of each vaccine for better and more comfortable care. 

To our knowledge, this is the first report detecting antibodies following the J&J one-shot COVID-19 vaccinations in a Hispanic and largely immigrant population. Our previous work has demonstrated that our population is largely a mono-lingual, uninsured, urban population employed as “essential workers” and therefore exempt from stay-at-home mandates. This population was severely impacted economically during the pandemic. Many people work in occupations that require travel for different job sites and therefore a single-shot vaccine option would be desirable. However, this population only represented 15% of the study population in the pivotal trial. Given that there was no detectable antibody present in 33% of our subjects (9 of 27) tested 30–120 days after the vaccine, this suggests that there might be a high rate of re-infection of Hispanic patients following the vaccine than other demographic populations. 

Although seroconversion and concurrent protection against any pathogen depend on several host factors, a median protective effect of over 5 months was observed in this study, confirming previous studies [67,68,69]. It is likely that the number of neutralizing antibodies also declines [70,71].

### Limitations

Limitations of this study include a small and heterogeneous convenience sample and the absence of assays for neutralizing antibody, B-cell memory, and T-cell responses. The small sample size of patients was likely exacerbated by factors limiting appointment scheduling such as the availability of the clinic to match patient availability and absence from work. Furthermore, the sample size makes drawing conclusions difficult. For example, in Table 3, there is a positive correlation for adverse symptoms and antibodies in subcategories of patient visits; the correlations from subcategory “60–90 days” come from a sample of nine patients, which highlights the weight each patient has on certain correlations.

## 5. Conclusions

This study demonstrates the vulnerability of a low-income minority population to COVID-19. Socioeconomic factors exacerbate this vulnerability. In this study of a working-class Hispanic population living in a densely populated urban environment, the Johnson & Johnson vaccine had limited efficacy. Co-morbidities which may impact vaccine efficacy include diabetes, hypertension, hyperlipidemia, and rheumatological diseases. Patients can be easily and efficiently monitored to determine the success and durability of COVID-19 vaccines using simple low-cost spike targeted COVID-19 antibody lateral flow devices to serve as an adjunct to assist community-based physicians to assess the health status of their patients. Further research of minority low-income urban populations is needed to better understand COVID-19 health care threats and how vulnerable population patient care can be optimized.

## Figures and Tables

**Table 1 vaccines-11-00148-t001:** List of subjects by ID number, the date of vaccination, and the 1st and 2nd follow-up appointment results of their antibody response.

ID Number (N = 74)	Vaccination Date (mm/dd/yyyy)	1st Antibody Result	2nd Antibody Result
1	6/8/2021	Positive	Positive
4	6/8/2021	Negative	Positive
5	6/8/2021	Positive	Positive
6	6/9/2021	Negative	Negative
7	6/9/2021	Positive	Positive
8	6/9/2021	Positive	Positive
9	6/9/2021	Positive	Positive
12	6/10/2021	Positive	Positive
14	6/10/2021	Positive	Positive
17	6/15/2021	Negative	Negative
23	6/18/2021	Positive	Positive
24	6/18/2021	Positive	Positive
25	6/18/2021	Positive	Positive
31	6/21/2021	Positive	Positive
34	6/21/2021	Negative	Negative
37	6/24/2021	Negative	Negative
39	6/26/2021	Negative	Negative
46	6/29/2021	Positive	Positive
54	7/12/2021	Positive	Positive
59	7/23/2021	Positive	Positive
67	8/28/2021	Positive	Positive
10	6/9/2021	Negative	
16	6/15/2021	Positive	
38	6/26/2021	Positive	
40	6/26/2021	Positive	
65	8/21/2021	Positive	
66	8/21/2021	Positive	
72	10/25/2021	Positive	
73	10/25/2021	Negative	
74	10/25/2021	Positive	
11	6/9/2021		Negative
18	6/15/2021		Negative
21	6/17/2021		Negative
22	6/17/2021		Negative
29	6/21/2021		Negative
33	6/21/2021		Negative
36	6/22/2021		Negative
43	6/28/2021		Negative
45	6/29/2021		Positive
15	6/11/2021		
47	6/29/2021		
48	6/29/2021		
53	7/12/2021		
55	7/13/2021		
56	7/14/2021		
57	7/16/2021		
61	7/28/2021		
63	8/3/2021		
64	8/3/2021		
68	8/30/2021		
69	8/30/2021		
70	8/30/2021		
71	8/30/2021		
2 *	6/8/2021		
3 *	6/8/2021		
13 *	6/10/2021		
19 *	6/15/2021		
20 *	6/15/2021		
26 *	6/19/2021		
27 *	6/19/2021		
28 *	6/19/2021		
30 *	6/21/2021		
32 *	6/21/2021		
35 *	6/22/2021		
41 *	6/26/2021		
42 *	6/28/2021		
44 *	6/28/2021		
49 *	7/6/2021		
50 *	7/6/2021		
51 *	7/6/2021		
52 *	7/6/2021		
58 *	7/16/2021		
60 *	Not vaccinated		
62 *	Not vaccinated		

* excluded from the study.

**Table 2 vaccines-11-00148-t002:** Summary of patient visits following vaccination.

Days Following Vaccination	Negative	Positive	Total
<30	4 (27%)	11 (73%)	15
30–60	1 (17%)	5 (83%)	6
60–90	3 (33%)	6 (67%)	9
90–120	6 (43%)	8 (57%)	14
>120	7 (44%)	9 (56%)	16

**Table 3 vaccines-11-00148-t003:** Correlation table between antibody results (negative or positive) and the variables.

Variables Correlated to Antibody Response	All Follow-Up Visits	Follow-Up Visit #1	Follow-Up Visit #2	<30 Days	30–60 Days	60–90 Days	90–120 Days	>120 Days
* Sample Size (N)	60 **	30	30	15	6	9	14	16
Days after Vaccination	−0.22	−0.03	−0.24	0.21	−0.31	−0.23	−0.44	−0.37
Adverse events:								
Pain at injection site	0.05	0.03	−0.21	0.12	0.32	−0.50		−0.29
Headache	0.05	−0.02		−0.12		0.25		
Fatigue	−0.23	−0.49		−0.85	−1.00	0.38		
Muscle aches	−0.02	−0.15		−0.34		0.25		
Nausea	0.01	−0.05		−0.44	0.20	0.25		
Dizziness	0.01	−0.05		−0.21	0.20			
Fever	−0.14	−0.14		−0.21				
Duration of symptoms (days)	0.03	−0.09	−0.21	−0.26		0.32		−0.29
Demographic data:								
Gender (0 = F, 1 = M)	−0.15	−0.07	−0.22	−0.34	0.45	0.19	−0.29	−0.16
Age	−0.12	−0.13	−0.12	−0.40	−0.07	0.15	−0.24	0.01
Diabetes	−0.08	−0.05	−0.07	−0.21	0.20		0.24	−0.22
Hypertension	−0.08	−0.21	0.03	−0.45	0.20		−0.06	0.10
Hyperlipidemia	−0.10	−0.14	−0.05	−0.21			0.24	−0.22
Rheumatological diseases	−0.08	−0.14	−0.04		0.20	−0.50		−0.05
Height	−0.14	−0.04	−0.24	−0.35	−0.15	0.01	−0.32	−0.28
Weight	−0.13	−0.05	−0.19	−0.55	0.34	−0.01	−0.27	−0.18

* Values are for sample size, not correlation. ** 53 unique participants vaccinated on the initial visit. However, 39 unique patients returned for one or both follow-up visits. 16 of the 39 did not have an antibody response in at least 1 follow-up. Notes: Follow-up visit #1 occurred from 0 to 90 days after vaccination. Visit #2 occurred after 90 days. Values equal to or greater than |0.3| are considered moderately correlated, whereas values equal to or greater than |0.5| are considered strongly correlated.

## Data Availability

Detailed data are available on request from the corresponding author.

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
