# Peer review of "Evaluating Johnson and Johnson COVID-19 Vaccination Outcomes in a Low-Income Hispanic Population"

_vaccines, 2023, doi:10.3390/vaccines11010148_

Round 1
Reviewer 1 Report
The manuscript “Evaluating Johnson and Johnson COVID-19 Vaccination Out-comes in a Low-Income Hispanic Population” assessed antibodies resulting from the J&J COVID-19 vaccine in Hispanic patient population. The study is important as the first study to assess antibodies following the J&J one-shot COVID 19 vaccinations in a population.
The work is interesting and precedent, although it is preliminary.
I think it is acceptable after some revision, taking into account the following points.
Minor points:
1. In table 1, vaccine date for # 60 and 62 are missing.
2. Line 234 “duration of symptoms (0.36).”, in the table 3, the value is 0.32; Line 244 “hypertension (0.13).” in the table 3, the value is 0.10.
3. The reference format is not consistent. For example, compare ref 16 and 21, the title is italicized or not; the page numbers are included or not.
Author Response
Please find your comments below with the revisions in red. Thank you again for highlighting the mistakes for correction.
- In table 1, vaccine date for # 60 and 62 are missing.
- Clarified that the patients were not vaccinated and do not have vaccination dates.
- Line 234 “duration of symptoms (0.36).”, in the table 3, the value is 0.32; Line 244 “hypertension (0.13).” in the table 3, the value is 0.10.
- Revised text to match the table’s results.
- The reference format is not consistent. For example, compare ref 16 and 21, the title is italicized or not; the page numbers are included or not.
- Revised References using the designated MDPI EndNote style from : https://endnote.com/style_download/mdpi/
Reviewer 2 Report
The manuscript describes a pilot clinical study for the evaluation of Johnson & Johnson Ad26.COV2.S COVID-19 vaccination outcome in a Low-Income Hispanic population using binding Antibodies measured using the Clungene® SARS-CoV-2 IgG/IgM Rapid 11 Test Cassette, which is a simple low-cost spike targeted COVID-19 antibody lateral flow device. The data provides novel insights for success and durability of COVID-19 vaccines in low-income minority populations and relationship between the binding antibody response and demographic and health status variables. However, the major limitation of the study is that the associations and conclusions are derived from a small heterogenous population based on the 39 participants that returned for at least one-follow-up appointment to measure antibodies. While the data is informative, more studies will be needed to confirm these results.
Few comments/questions for the authors are as follows:
1) Several of the vaccine adverse effects like Pain at injection site, Headache, Fatigue, Muscle aches, Nausea, Dizziness, Fever, Duration of symptoms etc. correlate negatively with the Antibody levels at the early time points but positively at the later (beyond 30 days) time points. What would be the possible explanation or related Ad26 vaccine mode of action for that? Does the observed reactogenicity affect expression of the Spike antigen? Are similar observations reported for other J&J vaccine clinical data for other populations as well?
2) There is one exception for the above which is for the pain at injection site that starts to correlate negatively (-0.5) at later time pints (>60 days) instead. What could be the possible explanation for that observation.
3) Is there race stratified data available from other J&J COVID-19 vaccine clinical trials where Hispanic populations from other geographical regions would have been included. If yes, are similar observations observed in those data as well.
4) The format of references is a bit inconsistent e.g., use of italics, page numbers etc. is not uniform that could be improved.
Author Response
Thanks you for your feedback that helped us mend some gaps in the paper. Please find your comments below in black and our revisions/comments in red.
-
Several of the vaccine adverse effects [… i.e. pain…] correlate negatively with the Antibody levels at the early time points but positively at the later (beyond 30 days) time points.
-
[1] What would be the possible explanation or related Ad26 vaccine mode of action for that?
-
[1-2] The study was not designed to research the mode of action for the vaccine, which may explain the observed pattern . Difficult to draw conclusions due to small sample size. Added to the limitations a statement openly addressing the issue [284-287].
-
-
[2] Does the observed reactogenicity affect expression of the Spike antigen?
- see above
-
[3] Are similar observations reported for other J&J vaccine clinical data for other populations as well?
- [3] Scarce reports for differences in vaccine effectiveness between ethnicity groups. We added our observations in the manuscript in the discussion. Lines 254-264 (search “Shay et al”)
-
- There is one exception for the above which is for the pain at injection site that starts to correlate negatively (-0.5) at later time pints (>60 days) instead. What could be the possible explanation for that observation.
- The data is simply skewed in the subcategories due to extremely small sample sizes. Also, the correlations aren’t summarizing the detailed results between people with an antibody response vs no response (aggregating everything); for example, in the 60-90 days group (n=9), only 1 person had injection site pain and it turned out they were also 1 of 3 people in the group without antibodies. Therefore, patients without antibodies appear to have a strong chance of pain at their injection site 60+ days after vaccination.
- Is there race stratified data available from other J&J COVID-19 vaccine clinical trials where Hispanic populations from other geographical regions would have been included. If yes, are similar observations observed in those data as well.
-
Addressed with additions mention in comment 1-[3] above
-
- The format of references is a bit inconsistent e.g., use of italics, page numbers etc. is not uniform that could be improved.
- Revised References using the designated MDPI EndNote style from: endnote.com/style_download/mdpi/